# DATASET DISTILLATION VIA ADVERSARIAL PREDICTION MATCHING

## ABSTRACT

Dataset distillation is the technique of synthesizing smaller condensed datasets from large original datasets while retaining necessary information to persist the effect. In this paper, we approach the dataset distillation problem from a novel perspective: we regard minimizing the prediction discrepancy on the real data distribution between models, which are respectively trained on the large original dataset and on the small distilled dataset, as a conduit for condensing information from the raw data into the distilled version. An adversarial framework is proposed to solve the problem efficiently. In contrast to existing distillation methods involving nested optimization or long-range gradient unrolling, our approach hinges on single-level optimization. This ensures the memory efficiency of our method and provides a flexible tradeoff between time and memory budgets, allowing us to distil ImageNet-1K using a minimum of only 6.5GB of GPU memory. Under the optimal tradeoff strategy, it requires only $2.5\times$ less memory and $5\times$ less runtime compared to the state-of-the-art. Empirically, our method can produce synthetic datasets just 10% the size of the original, yet achieve, on average, 94% of the test accuracy of models trained on the full original datasets including ImageNet-1K, significantly surpassing state-of-the-art. Additionally, extensive tests reveal that our distilled datasets excel in cross-architecture generalization capabilities.

## 1 INTRODUCTION

Cutting-edge machine learning models across diverse domains are progressively dependent on extensive datasets. Nevertheless, the enormity of these datasets introduces formidable challenges in aspects such as data storage, preprocessing, model training, continual learning (Aljundi et al., 2019a; Zhao & Bilen, 2021a), and other auxiliary tasks including hyperparameter tuning (Bergstra et al., 2011; Baik et al., 2020) and architecture search (Pham et al., 2018; Jin et al., 2019). Exploring surrogate datasets emerges as a viable solution, offering comparable efficacy while significantly reducing scale. While existing research on core-set selection has underscored the feasibility of employing representative subsets from the original datasets (Chen et al., 2010; Rebuffi et al., 2017; Aljundi et al., 2019b; Toneva et al., 2019), Dataset Distillation (DD) is a nascent approach that generate *new* data instances not present in the original sets. This methodology notably expands the scope for optimization, potentially retaining optimal training signals from the principal dataset, and has demonstrated remarkable efficacy and effectiveness [1].

Dataset distillation via imitating the behaviour of "teacher models" acquired through training on original datasets, by the intermediary "student models" trained on distilled datasets is perceived as an effective solution (Zhao & Bilen, 2021a; Cazenavette et al., 2022). These methods utilize the teacher-student paradigm as a medium to condense knowledge from raw data into a more succinct form. Particularly, Matching Training Trajectory (MTT) frameworks (Du et al., 2023; Cui et al., 2023) exhibit superior performance over alternate distillation paradigms. Compared to its predecessor, i.e., the gradient matching framework (Zhao et al., 2021), its superior performance can be attributed to the strategy of matching long-term training trajectories of teachers rather than short-term gradients. However, over-fitting on the synthetic dataset can cause a remarkable disparity between the obtained and real trajectories, deteriorating the effectiveness and incurring considerable mem-

---

[1]https://github.com/Guang000/Awesome-Dataset-Distillation

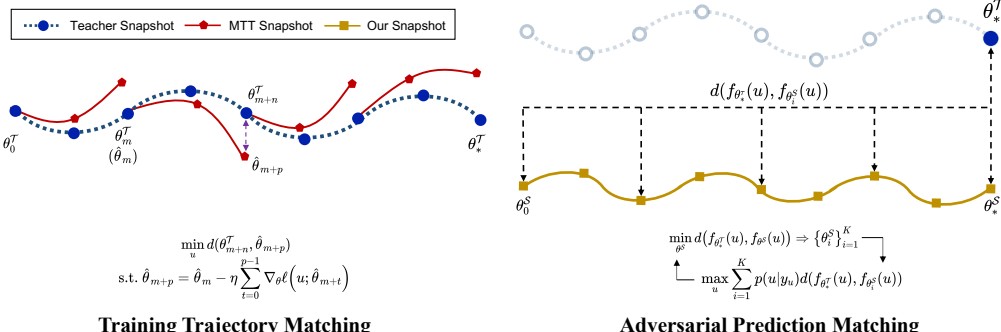

Figure 1: We contrast our adversarial prediction matching framework with the training trajectory matching (MTT) framework (Cazenavette et al., 2022). Our method has three primary advantages: *1.* It imitates the converged teachers' prediction, avoiding short-sightedness on local trajectories; *2.* Synthetic samples $u$ are updated by a single-level loss function, significantly enhancing memory complexity; *3.* It requires only one well-trained teacher, markedly diminishing storage overhead.

ory burdens during gradient derivation w.r.t synthetic samples. Hence, such methodologies remain confined to a localized imitation of behaviours elicited by teachers.

Prompted by these insights, we formally formulate a novel prediction-matching-based objective. This objective motivates synthetic data to enable student models to directly emulate the predictions of teacher models on the authentic data distributions of the original datasets, circumventing the risk of the distilled features becoming shortsighted. To effectively optimize the objective and bypass the nested optimization challenges in various prevailing methods (Du et al., 2023; Loo et al., 2023; Cui et al., 2023), we formulate an adversarial framework. A depiction of the basic proposition of our adversarial framework and its superiorities over the trajectory matching paradigm are provided in Figure 1. Intuitively, we obtain training snapshots of students by aligning predictions with teachers' on synthetic samples, then adversarially refine these samples to approach "critical points" in the real data distribution, which can cause substantial prediction disagreement between teachers and the aggregated snapshots with high probabilities in the original distribution. This strategy adeptly enables the distilled data to encapsulate crucial features from the original, motivating consistent alignment between intermediary students and teachers throughout the training phase.

Notably, although the utilized teacher-student paradigm in DD has a similar form to Knowledge Distillation (KD) (Hinton et al., 2015; Fang et al., 2019), especially our method hinges a prediction-matching paradigm, the research focal points are orthogonal. Instead of matching predictions between **two designated models**, the goal of DD is to create synthetic datasets endowed with prominent generalization ability that can off-the-shelf train unseen models. To this end, we devote ourselves to harnessing adequate information for the distilled data from the original, employing the prediction-matching paradigm as a conduit.

The contributions of this paper can be summarized as:

- A novel imitation-based prediction-matching notation is introduced for effective knowledge condensation from original datasets to distilled counterparts by aligning predictions of models, trained respectively on original and distilled datasets, over the authentic data distribution. An adversarial framework is consequently proposed to formulate the problem as a min-max game, allowing for trivial single-level optimization of distilled samples.

- Our method is proven to exhibit low memory complexity, offering a flexible tradeoff between memory consumption and runtime. Empirically, it enables the distillation of the ImageNet-1K dataset using a mere 6.5GB of GPU memory, achieving superior outcomes with only 2.5× less memory and 5× reduced runtime compared to the foremost approach.

- Experimental results illustrate that our distilled datasets, with just 10% the size of the original, can averagely attain 94% of the test accuracies of models than training on the original benchmark datasets, significantly surpassing existing solutions. Additionally, the distilled

dataset demonstrates superior performance in cross-architecture generalization and excels in the Neural Architecture Search (NAS) task.

## 2 RELATED WORK OF DATASET DISTILLATION

**Solving Bi-Level Optimizations.** A common approach in dataset distillation is to model the task as a bi-level optimization problem, aiming to minimize the generalization error on the original data caused by models trained on the distilled data (Wang et al., 2018; Bohdal et al., 2020; Zhou et al., 2022; Nguyen et al., 2021; Zhao & Bilen, 2021a; Loo et al., 2023). However, solving this problem is non-trivial, especially when the employed proxy models are optimized by gradient descent because it requires unrolling a complicated computational graph. To tackle this challenge, several recent works propose to approximate the model training with Kernel Ridge Regression (KRR) to get a closed-form solution for the optimum weights (Zhou et al., 2022; Nguyen et al., 2021) or directly solving the problem by the Implicit Gradient (IG) through convexifying the model training by the Neural Tangent Kernel (NTK) (Loo et al., 2023). Nonetheless, these methods either require extensive computations or are compromised by the looseness in the convex relaxation.

**Imitation-based Approaches.** Another widespread notion in dataset distillation involves mimicking specific behaviours elicited by the original dataset with those of the distilled dataset. Following this concept, the gradient matching frameworks (Zhao et al., 2021; Zhao & Bilen, 2021a; Kim et al., 2022) and training trajectory matching frameworks (Cazenavette et al., 2022; Du et al., 2023) propose to emulate the training dynamics of models experienced on the original dataset, making models trained on the distilled data converge to a neighbourhood in the parameter space. Although a significant improvement has been achieved by state-of-the-art methods, the update of synthetic data is typically memory-intensive as it also requires unrolling the computation graph of multistep gradient descent. While Cui et al. (2023) addresses this issue by allowing constant memory usage during gradient derivation, aligning trajectories obtained from extensive steps over synthetic data with long-term real trajectories becomes challenging. Consequent tradeoff also leads to inevitable truncation bias (Zhou et al., 2022). Additionally, distribution matching methods (Zhao & Bilen, 2021b; Wang et al., 2022) heuristically propose to align the feature distribution of original data with that of distilled data output by several proxy extractors, exhibiting inferior effectiveness.

## 3 METHOD

### 3.1 DATASET DISTILLATION VIA MATCHING PREDICTION ON REAL DATA DISTRIBUTION

Suppose a large labelled training dataset $\mathcal{T} = \{(x_i, y_i)\}_{i=1}^{|\mathcal{T}|}$, where $x_i \in \mathbb{R}^d$ and $y_i \in \{1, 2, \ldots, C\}$, is given to be distilled into a smaller distilled dataset $\mathcal{S} = \{(u_i, v_i)\}_{i=1}^{|\mathcal{S}|}$, with $u_i \in \mathbb{R}^d$ and $v_i \in \mathbb{R}^C$, such that $|\mathcal{S}| \ll |\mathcal{T}|$. Our objective is to construct $\mathcal{S}$ such that, for a given model architecture parameterized by $\theta \sim p(\theta)$, models trained respectively on $\mathcal{S}$ and $\mathcal{T}$ have the minimum discrepancy when predicting over the original data distribution with $p(x) = \sum_y p(x|y)p(y)$:

$$\min_{\mathcal{S}} \sum_{y=1}^{C} \int p(x|y) \cdot d\left(f_{\theta^{\mathcal{T}}}(x), f_{\theta^{\mathcal{S}}}(x)\right) dx \tag{1}$$

We omit $p(y)$ as we assume it follows a uniform distribution. Where, $d(.,.)$ refers to a distance function and $f_\theta(.)$ refers to the logtis output by the penultimate layer of models before the softmax. We employ logits for the prediction matching instead of the confidences normalized by the softmax function because logits contain more information for knowledge transfer (Yuan et al., 2020; Fu et al., 2020; Yin et al., 2020). $\theta^{\mathcal{T}}$ and $\theta^{\mathcal{S}}$ refer to the optimized parameters obtained by minimizing empirical loss over $\mathcal{T}$ and $\mathcal{S}$ respectively. Typically, this process is performed by an iterative optimization algorithm, denoted as $Alg$, such as gradient descent. For instance, $\theta^{\mathcal{S}} = Alg(\ell(\theta, \mathcal{S}), E)$, where $\ell(.,.)$ is any suitable loss function for model training, and $E$ denotes the number of training epochs.

We emphasize that minimizing the objective in Equation (1) is non-trivial as two reasons: *i)* calculating the gradient w.r.t $\mathcal{S}$ requires unrolling the recursive computation graph through the entire training process of $\theta^{\mathcal{S}}$, which is computationally expensive and memory-intensive, even $|\mathcal{S}|$ is small;

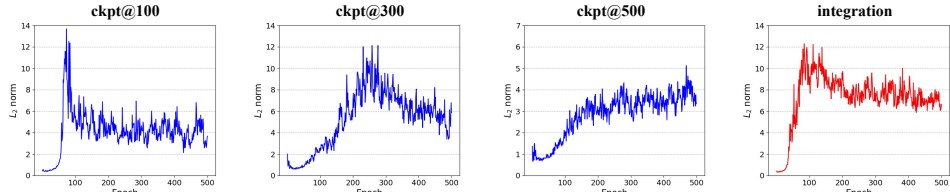

Figure 2: Trends of $L_2$ norm of $\nabla_\theta \ell(\theta, \mathcal{S})$ with $\mathcal{S}$ derived by Algorithm 1, utilizing checkpoints from a specified epoch (left three figures) and assorted epochs (the far right figure) for CIFAR-10, where $|\mathcal{S}| = 10 \times 50$. For example, "ckpt@100" denotes $\mathcal{S}$ is synthesized by referencing student models solely at epoch 100; "integration" utilizes models from epochs 100, 200, 300, 400 and 500.

*ii*) Directly minimizing the prediction discrepancy across the entire distribution is computationally infeasible due to the curse of dimensionality and the continuous nature of the variables.

## 3.2 ADVERSARIAL PREDICTION MATCHING FRAMEWORK

To tackle the above challenges, we first posit the existence of a subset $\mathcal{R}_e \subseteq \mathbb{R}^d$ within the feasible domain of $x$, where every element $x_e \in \mathcal{R}_e$ satisfies the following conditions: *i*) it is associated with a high conditional probability $p(x_e|y)$ for its corresponding class $y$; and *ii*) it induces substantial logits differences between the $f_{\theta^\mathcal{T}}$ and $f_{\theta_e^\mathcal{S}}$, where $e \in \{0, \ldots, E-1\}$ refers to the index of student checkpoints collected at each epoch of running $Alg$. We conjecture that such elements can "dominate" the expectation of logits difference over $p(x)$, denoted as:

$$\mathbb{E}_{x_e \in \mathcal{R}_e}[d\left(f_{\theta^\mathcal{T}}(x_e), f_{\theta_e^\mathcal{S}}(x_e)\right)] \geq \mathbb{E}_{x \sim p(x)}[d\left(f_{\theta^\mathcal{T}}(x), f_{\theta_e^\mathcal{S}}(x)\right)] \tag{2}$$

Namely, directly minimizing the expectation of logits difference over $x_e$ can facilitate the prediction alignment of each $\theta_e^\mathcal{S}$ to the target $\theta^\mathcal{T}$ over the original distribution. Based on this assumption, we propose to optimize the original objective by a curriculum-based adversarial framework. In each round, we first train a student model from $\theta_0^\mathcal{S}$ by the optimizer $Alg$ on $\mathcal{S} = \{(u_i, f_{\theta^\mathcal{T}}(u_i))\}$ for $E$ epochs with $d(.,.)$ as the loss function, then alternately force synthetic samples $\{u_i\}_{i=1}^{|\mathcal{S}|}$ to approach "hard samples" $x_e \in \mathcal{R}_e, \forall e \in \{0, \ldots, E-1\}$ with the following target:

$$\max_u \frac{1}{|\mathcal{S}|} \sum_{i=1}^{|\mathcal{S}|} \sum_{e=0}^{E-1} \log[p(u_i|y_{u_i})d\left(f_{\theta^\mathcal{T}}(u_i), f_{\theta_e^\mathcal{S}}(u_i)\right)] \tag{3}$$

Where, $y_{u_i}$ refers to the class in which $u_i$ is associated for distilling. However, the inherent complexity in estimating $p(u|y_u)$ due to the high-dimensional nature of data still hinders the optimization of Equation (3). To address this issue, we advocate for substituting $p(u|y_u)$ with $p(y_u|u)$ for the maximization of the stated target. According to Bayes' theorem, we have $p(u|y_u) \propto p(y_u|u)p(u)$, presuming a uniform distribution for $p(y)$. Given the stark contrast between the high-dimensional continuum inherent in $u$ and the finite discreteness in $y$, we posit, heuristically, that prioritizing the maximization of $p(y_u|u)$ is a plausible strategy for the overall maximization of $p(y_u|u)p(u)$. Regarding the teacher model derived from the original dataset is reliable, we employ $p(y_u|u; \theta^\mathcal{T}) = \text{softmax}(f_{\theta^\mathcal{T}}(u))_{y_u}$ as an approximation for $p(y_u|u)$. Consequently, our loss function designed for updating $\{u_i\}_{i=1}^{|\mathcal{S}|}$ to maximize the stated target is defined as follows:

$$\mathcal{L}_u = \frac{1}{|\mathcal{S}|} \sum_{i=1}^{|\mathcal{S}|} \sum_{e=0}^{E-1} -\log\left[d\left(f_{\theta^\mathcal{T}}(u_i), f_{\theta_e^\mathcal{S}}(u_i)\right)\right] + \alpha H\left(y_{u_i}, p(y_{u_i}|u_i; \theta^\mathcal{T})\right) \tag{4}$$

Where, $H\left(y_{u_i}, p(y_{u_i}|u_i; \theta^\mathcal{T})\right) = -\sum_{c=1}^C y_c \log\left(p(y_c|u_i; \theta^\mathcal{T})\right) = -\log\left(p(y_{u_i}|u_i; \theta^\mathcal{T})\right)$ denotes the cross-entropy loss of $\theta^\mathcal{T}$ on $u_i$ w.r.t the associated distilling class $y_{u_i}$ and $\alpha$ refers to a hyperparameter for flexible tradeoff. Intuitively, this loss function encourages the synthetic samples to evolve into valuable, hard points, thus contributing beneficially to the overall training process of $\theta^\mathcal{S}$.

**Practical Implementation.** We outline the overall procedure of our proposed Adversarial Prediction Matching (APM) framework in Algorithm 1. We initialize $\mathcal{S}$ by sampling real data from

---

**Algorithm 1:** Dataset distillation via adversarial prediction matching

---

**Input:** An architecture parameterized by $\theta$; A set of well-trained teacher weights $\{\theta^{\mathcal{T}}\}$;
  Rounds $R$; Epochs $E$; Number of select checkpoints $K$; Model learning rate $\eta$;
  Synthetic samples learning rate $\gamma$; Mini-batch size $B$

**Output:** Distilled synthetic dataset $\mathcal{S}$

1 Initialize synthetic samples by randomly sampling $\{u_i\}_{i=1}^{|\mathcal{S}|} \subset \{x_i\}_{i=1}^{|\mathcal{T}|}$;

2 **for** $r = 1$ **to** $R$ **do**

3      Initialize $\theta^{\mathcal{T}} \sim \{\theta^{\mathcal{T}}\}$, $\theta_0^{\mathcal{S}} \sim p(\theta)$, and an empty checkpoint list;

4      Randomly sample a mini-batch $\{u_i^r\}_{i=1}^B \subseteq \{u_i\}_{i=1}^{|\mathcal{S}|}$;

5      **for** $e = 1$ **to** $E$ **do**

6          Calculate the prediction matching loss: $\mathcal{L}_\theta = \frac{1}{B}\sum_{i=1}^B d(f_{\theta^{\mathcal{T}}}(u_i^r), f_{\theta_{e-1}^{\mathcal{S}}}(u_i^r))$;

7          Update $\theta_e^{\mathcal{S}} = \theta_{e-1}^{\mathcal{S}} - \eta \cdot \nabla_{\theta_{e-1}^{\mathcal{S}}} \mathcal{L}_\theta$;

8          **if** $e \mod \lfloor E/K \rfloor = 0$ **then**

9              Store $\theta_e^{\mathcal{S}}$ in the checkpoint list;

10      Calculate $\mathcal{L}_u$ in Equation (4) with $K$ checkpoints;

11      Update $u^r = u^r - \gamma \cdot \nabla_{u^r} \mathcal{L}_u$;

12 Calculate $v_i = f_{\theta^{\mathcal{T}}}(u_i), \forall u_i$ and set $\mathcal{S} = \{(u_i, v_i)\}_{i=1}^{|\mathcal{S}|}$;

---

$\mathcal{T}$, ensuring each $u_i$ has a high initial probability of $p(y_{u_i}|u_i; \theta^{\mathcal{T}})$. Notably, to fulfil the inherent generalization requisite, we adopt the strategy from MTT frameworks (Cazenavette et al., 2022; Du et al., 2023; Cui et al., 2023), obtaining a set of teacher models, $\{\theta^{\mathcal{T}}\}$, prior to commencement. Subsequently, in each iteration, a $\theta^{\mathcal{T}}$ is sampled from the set to be aligned with a newly initialized $\theta_0^{\mathcal{S}} \sim p(\theta)$. To constrain the memory overhead, we compute the loss in Equation (4) with $K$ select checkpoints, collected at regular intervals during training. We empirically validate in Section 4.4 that this does not compromise the efficacy of our method.

**Necessity of Involving Training Snapshots in $\mathcal{L}_u$.** Ideally, a distilled dataset should remain informative throughout the entire model training process, avoiding limitations to specific stages. Nonetheless, our observations indicate that relying on proxy student models from a singular stage can introduce a significant inductive bias. To clarify this, we generate synthetic samples using Algorithm 1 but with checkpoints from a specified epoch to calculate $\mathcal{L}_u$. Then, the resultant dataset is used to train newly initialized models, and variations in gradient norm relative to model parameters are observed. Figure 2 reveals a noteworthy trend where peaks in gradient norms are prominent and only appear near the epochs of the selected checkpoints. Conversely, jointly utilizing these checkpoints for synthetic sample generation exhibits a stable pattern throughout training.

### 3.3 Superiority Analysis

**Low Memory Complexity.** Based on the definition of the loss function proposed for updating synthetic samples in Equation (4), all computational operations conducted on each synthetic sample $u_i$, e.g., the computation of each $d(f_{\theta^{\mathcal{T}}}(u_i), f_{\theta^{\mathcal{S}}}(u_i))$ and $H\left(y_{u_i}, p(y_{u_i}|u_i; \theta^{\mathcal{T}})\right)$ (abbreviated as $d_i^e$ and $H_i$, respectively), are integrated by addition. Thanks to this characteristic, two inherent advantages in memory efficiency emerge: *1.* the gradient of $\mathcal{L}_u$ w.r.t each $u_i$ can be independently derived without interference from other samples of $u_{j \neq i}$; *2.* the partial derivative of each computation term, e.g., $\frac{\partial d_i^e}{\partial u_i}$ or $\frac{\partial H_i}{\partial u_i}$, can be independently calculated, and finally updating $u_i$ based on the accumulated gradients of each term. Thus, the minimal memory complexity of our method can be expressed as $\mathcal{O}(\mathcal{G}_{\text{ours}})$, where $\mathcal{G}_{ours}$ refers to the computation graph size for a single input sample, and it is dominated by $\frac{\partial d_i^e}{\partial u_i}$ or $\frac{\partial H_i}{\partial u_i}$ which only involves a trivial backpropagation through models. In contrast, TESLA (Cui et al., 2023) achieves the optimal memory efficiency among MMT frameworks by reducing the memory complexity of the original (Cazenavette et al., 2022) from $\mathcal{O}(TB\mathcal{G}_{MTT})$ to $\mathcal{O}(B\mathcal{G}_{MTT})$, where $T$ and $B$ denote specified updating steps of deriving trajectories and the mini-batch updating size for synthetic data, respectively. Compared with ours, except the mini-batch size $B$, the computation graph size $\mathcal{G}_{MTT}$ is dominated by a complicated Jacobian

matrix $\frac{\partial}{\partial u_i} \nabla_\theta \ell(\theta, \mathcal{S}) \in \mathbb{R}^{|\theta| \times d}$. The remarkable memory efficiency of our method enables the distillation of various-size synthetic datasets for large-scale datasets, such as ImageNet-1K, even under memory-constrained circumstances.

**Customized Tradeoff between Time and Memory.**  We show that $\nabla_{u_i} \mathcal{L}_u$ for each $u_i$ can be calculated independently and further decomposed into partial derivatives of each computational term. In practice, this implies multiple model invocations for separate calculations, elongating the runtime. Thus, our algorithm optionally allows computation of the gradients w.r.t multiple $u$ simultaneously, offering a flexible tradeoff between memory and runtime. Further, Section 4.3 empirically validates our capability to distribute distillation tasks to independent workers (e.g., generating $N$ subsets of size $|\mathcal{S}|/N$ in parallel and aggregating), a feature absent in prior dataset distillation methods.

**Significantly Reduced Storage vs. MTT.**  Our approach only requires matching predictions of the converged teacher, requiring one checkpoint per teacher. However, MTT frameworks necessitate saving checkpoints from every epoch of training a teacher. Therefore, even utilizing the same setup to acquire an equal number of teachers, the storage usage of our method is only a fraction (i.e., 1/epoch) of that of MTT frameworks. Moreover, an ablation study in Section 4.4 demonstrates that the performance of our method obtained with 10 teachers is comparable to that with 100 teachers.

## 4 EXPERIMENTS

### 4.1 EXPERIMENTAL SETUPS

**Datasets.**  We evaluate the performance of our method on distilling four benchmark datasets, including CIFAR-10/100 (Krizhevsky & Hinton, 2009), Tiny ImageNet (Le & Yang, 2015) and ImageNet-1K (Russakovsky et al., 2015). Notably, following the protocol of Zhou et al. (2022) and Cui et al. (2023), we resize the resolution of ImageNet-1K images to $64 \times 64$ in our evaluations for fair comparison. We provide further detailed information of benchmark datasets in Appendix A.1.

**Evaluation Setup and Baselines.**  We employ several recent distillation frameworks as baselines including a short-term gradient matching method Data Siamese Augmentation (DSA) (Zhao & Bilen, 2021a), a feature distribution matching method Distribution Matching (DM) (Zhao & Bilen, 2021b), two bi-level-optimization-based methods which are FRePo (Zhou et al., 2022) and RCIG (Loo et al., 2023), and three training trajectory matching methods which are Matching Training Trajectory (MTT) (Cazenavette et al., 2022) and its two variations TESLA (Cui et al., 2023) and Flat Trajectory Distillation (FTD) (Du et al., 2023).

**Evaluation Metrics.**  To evaluate compared methods, we adhere to the standard protocol prevalent in the existing literature. We train 5 randomly initialized networks from scratch over the distilled dataset and evaluate their performances by reporting the mean and standard deviation of their accuracy on the test set. We use the terminology "IPC" (an acronym for "images per class") in dataset distillation to indicate the size of distilled datasets. Specifically, we examine an IPC of 50/500/1000 for CIFAR-10 and an IPC of 10/50/500 for CIFAR-100, Tiny ImageNet, and ImageNet-1K.

**Implementation Details.**  We implement our method based on the procedure of Algorithm 1. Specifically, we set the teacher number of $\{\theta^T\}$ as 100 to be fairly compared with MTT and its variations TESLA and FTD. We set the distance metric $d$ by $L_1$ distance, the running round $R = 2,500$, the number of checkpoints $K = 5$ and the coefficient of cross-entropy regularizer $\alpha = 0.1$ by default. Further details of our hyper-parameter setup and guidelines are in Appendix A.7. For a fair comparison, we stay with all the compared precedents to adopt the 3-layer ConvNet architecture (Gidaris & Komodakis, 2018) for distilling CIFAR-10 and CIFAR-100, and extend to the 4-layer ConvNet for Tiny ImageNet and ImageNet-1K. Moreover, we also employ ZCA whitening for CIFAR-10 and CIFAR-100 as done by previous works (Zhou et al., 2022; Loo et al., 2023; Cazenavette et al., 2022; Du et al., 2023; Cui et al., 2023). All the experimental results of our method can be obtained on one RTX4090 GPU, including the extensive ImageNet-1K dataset.

### 4.2 COMPARISON TO STATE-OF-THE-ART

**Evaluation on ConvNets.**  As illustrated by Table 1, our method demonstrates superior performance than all the baselines among all the evaluation scenarios. Notably, ConvNet models trained

Table 1: Comparison of the test accuracy (%) between ours and other distillation methods on CIFAR10/100, Tiny ImageNet and ImageNet-1K datasets. Entries marked as absent are due to scalability issues. See Appendix A.2 for detailed reasons.

| Dataset | IPC | Random | DSA | DM | FRePo | RCIG | MTT | TESLA | FTD | **Ours** | Whole |
|---|---|---|---|---|---|---|---|---|---|---|---|
| CIFAR-10 | 50 | 50.2±0.5 | 60.6±0.5 | 63.0±0.4 | 71.7±0.2 | 73.5±0.3 | 71.6±0.2 | 72.6±0.7 | 73.8±0.2 | **75.0±0.2** | 85.6±0.2 |
| | 500 | 71.9±0.3 | - | 74.3±0.2 | - | - | 78.6±0.2 | 78.4±0.5 | 78.7±0.2 | **83.4±0.3** | |
| | 1000 | 78.2±0.3 | - | 79.2±0.2 | - | - | 79.7±0.2 | 80.3±0.3 | 81.2±0.2 | **84.6±0.2** | |
| CIFAR-100 | 10 | 14.4±0.5 | 32.3±0.3 | 29.7±0.3 | 42.5±0.2 | 44.1±0.4 | 40.1±0.4 | 41.7±0.3 | 43.4±0.3 | **44.6±0.3** | 56.8±0.2 |
| | 50 | 30.5±0.3 | 42.8±0.4 | 43.6±0.4 | 44.3±0.2 | 46.7±0.3 | 47.7±0.2 | 47.9±0.3 | 50.7±0.3 | **53.3±0.2** | |
| | 100 | 43.1±0.2 | 44.7±0.2 | 47.1±0.4 | - | 47.4±0.3 | 50.3±0.1 | 51.1±0.3 | 52.8±0.3 | **55.2±0.2** | |
| Tiny ImageNet | 10 | 4.8±0.3 | 15.9±0.2 | 12.9±0.4 | 25.4±0.2 | 29.4±0.2 | 23.2±0.2 | 14.1±0.3 | 24.5±0.2 | **30.0±0.3** | 40.0±0.4 |
| | 50 | 15.1±0.3 | 21.6±0.3 | 21.2±0.3 | - | - | 28.0±0.3 | 33.4±0.5 | 31.5±0.3 | **38.2±0.4** | |
| | 100 | 24.3±0.3 | - | 29.4±0.2 | - | - | 33.7±0.6 | 34.7±0.2 | 34.5±0.4 | **39.6±0.2** | |
| ImageNet-1K | 10 | 4.1±0.1 | - | - | - | - | - | 17.8±1.3 | - | **24.8±0.2** | 37.2±0.5 |
| | 50 | 16.2±0.8 | - | - | - | - | - | 27.9±1.2 | - | **30.7±0.3** | |
| | 100 | 19.5±0.5 | - | - | - | - | - | 29.2±1.0 | - | **32.6±0.3** | |

on our distilled datasets, which are just $10\%$ the size of the original, manage to achieve, on average, $96\%$ of the test accuracies of models trained on the full CIFAR-10, CIFAR-100, and Tiny ImageNet. Conversely, models trained on equivalent-sized random subsets can only reach $58\%$, while the state-of-the-art trajectory matching framework, FTD, achieves a maximum of $87\%$. Specifically, the average test accuracies of our method surpass those of FTD by $3.1\%$ on CIFAR-10, $2.1\%$ on CIFAR-100, and $5.8\%$ on Tiny ImageNet. Compared to the state-of-the-art bi-level optimization framework RCIG, our framework not only exhibits superior performances at a smaller IPC but also demonstrates feasibility and effectiveness when generating synthetic sets with a larger IPC. Furthermore, thanks to the memory efficiency of our adversarial prediction matching framework, our method stands as one of the only two capable of successfully distilling the extensive ImageNet-1K dataset, and it additionally exceeds the competitor, TESLA, by $4.4\%$ on average. A visualization study of our synthetic data is shown in Appendix A.8.

**Cross-Architecture Generalization.** In Table 2, we assess the generalization performance of datasets distilled by our method and by several representative baselines, including DSA, DM, RCIG, and FTD, on unseen model architectures over CIFAR and Tiny ImageNet with an IPC of 50. We utilize three different neural network architectures for evaluation: VGG11 (Simonyan & Zisserman, 2015), ResNet18 (He et al., 2016) and Vision Transformer (ViT) (Dosovitskiy et al., 2021). Those models are widely employed for cross-architecture evaluations of recent dataset distillation works (Cui et al., 2022; 2023; Cazenavette et al., 2023). The results demonstrate that models learned with our synthetic data have superior performances than models trained on datasets synthesized by other baselines. Additionally, compared to the results on ConvNet shown in Table 1, we surprisingly noticed larger performance improvements of our method to the state-of-the-art FTD method across multiple unseen models. Moreover, our results obtained on ResNet18 even exceed the results obtained on ConvNet when testing on CIFAR-100 and Tiny ImageNet.The remarkable transferability demonstrates that our method is more capable of capturing the essential features for learning. We also report the generalization result of our distilled dataset of ImageNet-1K in Appendix A.3.

### 4.3 MEMORY AND TIME EFFICIENCY ANALYSIS

**High Efficiency for Distilling ImageNet-1K.** To empirically evident the superiorities of the memory complexity and the flexible tradeoff between runtime and memory usage we discussed in Section 3.3, we compare our method with TESLA, which exhibits the optimal memory complexity among trajectory matching frameworks, for distilling ImageNet-1K with an IPC of 50. We conduct tests on a Tesla A100 GPU (80GB) for a fair comparison. Figure 3 represents the averaged peak memory usage and runtime per iteration over 50 rounds collected by running algorithms with the reported default hyper-parameters for obtaining the results shown in Table 1. Both methods set the mini-batch size for updating synthetic samples in each iteration as $500$. However, our method allows for an equivalent update by distributing the updates of 500 samples across multiple segments. For instance, "Segment-100" implies dividing 500 samples into five parts and updating sequentially. Through this approach, our method can flexibly tradeoff memory usage and runtime, enabling the distillation of ImageNet-1K with merely 6.5GB of memory and achieving superior distillation performance. Additionally, we observe that even when updating 500 samples at once, our method

Table 2: Cross-architecture generalization performance in test accuracy (%) of synthetic datasets distilled for CIFAR-10/100 and Tiny ImageNet at an IPC of 50.

| Dataset | Model | Random | DSA | DM | RCIG | FTD | **Ours** |
|---|---|---|---|---|---|---|---|
| CIFAR-10 | VGG11 | 46.7±0.6 | 51.0±1.1 | 59.2±0.8 | 47.9±0.3 | 59.1±0.2 | **65.8±1.0** |
| | ResNet18 | 48.1±0.3 | 47.3±1.0 | 57.5±0.9 | 59.5±0.2 | 64.7±0.3 | **69.2±0.8** |
| | ViT | 38.9±0.6 | 22.6±1.0 | 27.2±1.6 | 27.7±0.8 | 38.7±0.9 | **42.2±1.1** |
| CIFAR-100 | VGG11 | 28.4±0.3 | 29.8±0.1 | 34.6±0.4 | 36.7±0.1 | 42.5±0.2 | **46.6±0.2** |
| | ResNet18 | 41.0±0.8 | 41.7±0.7 | 37.7±0.5 | 36.5±0.4 | 48.4±0.1 | **55.2±0.3** |
| | ViT | 25.5±0.2 | 27.4±0.3 | 28.2±0.6 | 15.7±0.6 | 30.2±0.5 | **35.2±0.6** |
| Tiny ImageNet | VGG11 | 21.1±0.3 | 21.5±0.3 | 22.0±0.4 | - | 27.2±0.3 | **35.1±0.1** |
| | ResNet18 | 30.7±0.8 | 30.9±1.1 | 20.3±0.4 | - | 35.7±0.6 | **40.2±0.2** |
| | ViT | 17.4±0.6 | 18.6±0.3 | 18.1±0.4 | - | 21.5±0.1 | **24.5±0.8** |

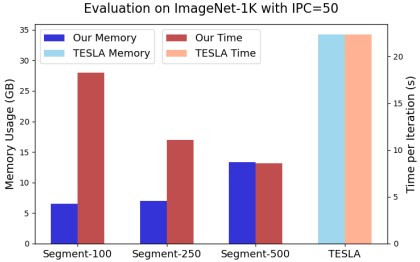

Figure 3: Flexible tradeoff between memory usage and runtime per iteration of our method compared with TESLA's for distilling ImageNet-1K with an IPC of 50.

Figure 4: Test accuracy (%) varying with merging different numbers of distributively generated sub-batches. IPC of each sub-batch is 50 for CIFAR-10 and 10 for Tiny ImageNet.

consumes $2.5\times$ less memory than Tesla. Given that the default running iteration of Tesla is double ours, our overall runtime is only one-fifth of theirs.

**Parallelizability of Dataset Distillation.** To further enhance productivity, a logical approach is to divide the generation of a synthetic dataset and allocate it to multiple workers for parallel execution without compromising effectiveness. We refer to this feature as the parallelizability of dataset distillation methods. We evaluate this feature of our method and several representative frameworks including DM, FRePo and MTT in Figure 4. Specifically, we separately craft 10 groups of synthetic data with an IPC of 50 for CIFAR-10 and 5 groups with an IPC of 10 for Tiny ImageNet. We can see from the figure that the performances of models attained on our synthetic datasets persistently increase with the growing number of merged groups. Moreover, we are the only framework that can achieve equivalent performances as the original results shown in Table 1. This indicates that, unlike other frameworks, running our method in parallel does not capture repetitive or redundant features.

## 4.4 ABLATION STUDY

We study the effectiveness or the influence of the following four components of our method. **Number of Pre-trained Teachers:** To enhance the generalization ability, we propose to match prediction with different teachers sampled from a pre-gathered pool $\{\theta^T\}$. Figure 5(a) reveals a consistent performance increase with the size of $\{\theta^T\}$, emphasizing the importance of dynamically altering the teacher model to circumvent overfitting to a singular model. Notably, the result with 10 teachers is nearly equivalent to that with 100 teachers, highlighting the potential for computation and storage resource conservation. **Number of Involved Checkpoints:** As illustrated by Figure 5(b), utilizing $K$ checkpoints in Equation (4) improves the performance of only using the last checkpoint. Besides, we observe that setting $K$ larger than 5 will not cause significant changes in the performance, allowing equivalent effectiveness with reduced resources. **Mini-Batch Size:** In Sec 3.3, we emphasize that the loss function for updating one synthetic sample is computationally independent of others. Fig 5(c) shows the invariance of performances of synthetic datasets distilled with different mini-batch sizes $B$. Thus, both memory consumption and runtime of each round can be effectively reduced when the mini-batch size is small. **Number of Epoch for the Model Update:** We exam-

Table 3: Comparison of Spearman's ranking correlation of NAS on CIFAR-10. Results are obtained from models trained on various distilled datasets with an IPC of 50 (i.e., 1% of the original dataset) over 200 epochs. For reference, the correlation achieved by training models on the full original dataset for 20 epochs stands at 0.52, albeit requiring $10\times$ the search time than ours.

| Random | DSA | DM | MTT | FRePo | **Ours** |
|--------|-----|-----|-----|-------|----------|
| 0.18±0.06 | 0.12±0.07 | 0.13±0.06 | 0.31±0.01 | -0.07±0.07 | **0.47±0.03** |

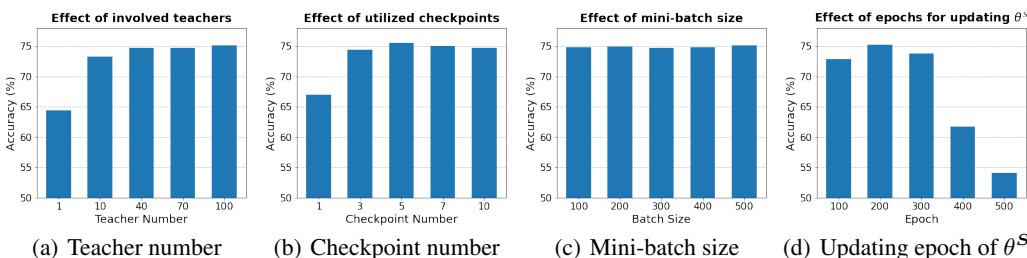

| (a) Teacher number | (b) Checkpoint number | (c) Mini-batch size | (d) Updating epoch of $\theta^S$ |
|---|---|---|---|

Figure 5: Ablation studies of our method conducted on CIFAR-10 with an IPC of 50.

ine the effect of the epoch number $E$ for training student models on the synthetic dataset in Figure 5(d). We can see a significant decrease when $E$ is set higher than 300. This is because a large $E$ will cause most of the $K$ collected checkpoints to be nearly converged. Thus, condensed features derived by using them as proxies lack sufficient knowledge beneficial for the early stages of model training. Moreover, we comprehensively study the influence of varying the coefficient of cross-entropy regularizer, $\alpha$, over the four benchmark datasets in Appendix A.5.

### 4.5 APPLICATION: NEURAL ARCHITECTURE SEARCH

To explore the practicality of our method, we utilize the distilled datasets as a proxy for the intricate Neural Architecture Search (NAS) task. For this, we adopt the same NAS-Bench-201 task (Dong & Yang, 2020), previously employed by a recent dataset distillation benchmark (Cui et al., 2022). Specifically, we randomly sample 100 networks from NAS-Bench-201, in which the search space is comprised of 15,625 networks with ground-truth performances collected by training on the entire training dataset of CIFAR-10 for 50 epochs under 5 random seeds, and ranked according to their average accuracy on a held-out validation set of 10k images. We compare our methods with various baselines including Random (a randomly selected subset), DSA, DM, MTT and FRePo by training the 100 different architectural models over synthetic datasets with an IPC of 50 for 200 epochs. The Spearman's correlation between the rankings derived from the synthetic datasets and the ground-truth ranking is used as the evaluation metric. Comparison results are reported in Table 3. Notably, due to the inherent variability in training, an average ranking correlation of 0.76 is achieved locally even when adhering to the ground-truth protocol. We can clearly see from the table that the ranking correlation attained by our method markedly surpasses other baselines. Especially, although MTT and FRePo's test accuracies in Table 1 are on par with ours for an IPC of 50 on CIFAR-10, the efficiency of our method for this NAS task is evident. Moreover, the correlation achieved by training models on the entire original training dataset of CIFAR-10 for 20 epochs is 0.52. However, our distilled datasets deliver comparable performance in just one-tenth of the search time.

## 5 CONCLUSIONS

In this work, we introduce an innovative formulation for dataset distillation via prediction matching, realized with an adversarial framework, devised for effective problem resolution. Comprehensive experiments validate that our distilled datasets surpass contemporary benchmarks, showcasing exceptional cross-architecture capabilities. Furthermore, we illustrate the superior memory efficiency and practical viability of large-scale dataset distillation of our method. For future research, we intend to explore advancements in efficiency by diminishing its reliance on the number of teacher models.

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
