# OpenReview forum: "Dataset Distillation via Adversarial Prediction Matching"
_ICLR.cc/2024/Conference — Submitted to ICLR 2024_

### Official Review · Reviewer_YDsU · 2023-10-30

**Soundness:** 3 good
**Presentation:** 3 good
**Contribution:** 2 fair
**Rating:** 5
**Confidence:** 4

**Summary:**

This paper introduces a novel approach to dataset distillation that focuses on minimizing the prediction discrepancy between models trained on original large datasets and their distilled counterparts. The authors propose an adversarial framework that employs single-level optimization, differing from traditional methods that use nested optimization or gradient unrolling. This new method is not only memory-efficient, requiring as little as 6.5GB of GPU memory to distill ImageNet-1K, but also reduces the memory and runtime needed by 2.5 and 5 times, respectively, compared to the latest techniques.

**Strengths:**

1. The memory issue and training cost are the two challenges of existing dataset distillation, which prohibit the application of dataset distillation in large-scale datasets. This paper proposes an efficient dataset distillation method for both memory and training costs to address the two challenges.

2. The improvement of this method is significant.

3. The paper is well-organized and easy to follow. The mathematical formulation and figures in Section 3 are good.

**Weaknesses:**

1. The adersarial prediction matching seems to be close to DD [1]. Could the authors summarize the differences and improvements compared to DD?

2. If the loss function in line 6 of algorithm 1 is based on batch, then why is the memory complexity in section 3.3 only the graph size? The batch size should be counted as well.

3. The authors append some visualizations of the distilled images in the appendix.
4. The authors should have more experiments on NAS. Because NAS is a practical application of dataset distillation, The experiments on Table 3 are not sufficient.

[1] Tongzhou Wang, Jun-Yan Zhu, Antonio Torralba, and Alexei A. Efros. Dataset distillation. CoRR,
abs/1811.10959, 2018.

**Questions:**

Some questions are stated as weaknesses. The other questions are listed below.

1. The study of memory cost presented in Figure 3 could be more detailed. For instance, a comparison of complexity versus batch size or number of steps relative to the baseline TESLA would be informative.

2. Visualization of the distilled ImageNet dataset.

3. There seems to be a performance gap between the CNNs and the ViTs. Could authors have the cross-archi experiments on the distilled images trained with ViT.  (Train on ViT and test on CNNs).

4. The SOTA of imageNet-1k should be [1] instead of TESLA. the claim of SOTA in imagenet-1k is not objective.

[2] Zeyuan Yin, Eric Xing, and Zhiqiang Shen. Squeeze, recover and relabel: Dataset condensation at
imagenet scale from a new perspective. In NeurIPS, 2023. 1, 2, 3, 4, 5,

---

> ### Author Response · Authors · 2023-11-16
> **Response to Reviewer YDsU (1/3)**
>
> Thank you for recognizing the strengths of our paper, particularly our approach to dataset distillation which effectively tackles the memory and training cost challenges in large-scale datasets. Your acknowledgement of the notable improvements our method provides is much appreciated. We also value your feedback on the organization of the paper and the clarity of Section 3. For the weaknesses and questions, you've raised, we would like to offer the following responses.
>
> **Weakness 1: The adversarial prediction matching seems to be close to DD. Could the authors summarize the differences and improvements compared to DD?**
>
> **Answer:**
>
> Since the reviewer did not provide a detailed explanation of why our method is close to DD, we attempt to **summarize the differences between our method and DD, as well as our improvements**, from **three key aspects**.
>
> First, **fundamentally, our problem formulation for solving the dataset distillation task differs from DD's proposed formulation**. Specifically, DD formulates the dataset distillation task as a bi-level optimization problem, aiming to minimize the generalization error on the original data caused by models trained on the distilled data. In contrast, we introduce an innovative imitation-based surrogate objective. This objective optimizes distilled data to enable proxy student models trained on it to directly emulate the predictions of proxy teacher models, trained on the original dataset, on the authentic data distributions.
>
> Second, **the adversarial strategy we introduced for effectively optimizing the formulated prediction-matching objective provides a novel intuition for synthetic samples, distinct from existing methods**. Specifically, the objective defined by Equation (3) encourages synthetic samples to capture informative features recognized by the teacher but absent in the current synthetic samples. This reduces the gap in contained features between the original and the distilled datasets.
>
> Third, **our optimization for the synthetic data relies on optimizing a single-level loss function defined in Equation (4) with remarkably low memory complexity**. In contrast, even if DD truncates the inner optimization of training models on the distilled data to only one-step gradient descent, it still requires a complex nested optimization when calculating the gradient for updating synthetic data.
>
> Finally, it is important to note that we did not employ DD as a baseline method in our paper. This decision was based on **the significant performance gap between DD and our method**, as well as **the high computational and memory complexity of DD, which limits its applicability in various scenarios we evaluated**.
>
> **Weakness 2: If the loss function in line 6 of algorithm 1 is based on batch, then why is the memory complexity in section 3.3 only the graph size? The batch size should be counted as well.**
>
> **Answer:**
>
> We appreciate your meticulous attention to detail and the opportunity to underscore one of our key contributions: a significant reduction in memory complexity compared to TESLA.
>
> As outlined in the first paragraph of Section 3.3, we explicitly highlighted that **all computational operations performed on each synthetic sample** in our loss function, as defined by Equation (4), **are consolidated through addition**. This unique characteristic allows us to **independently compute the gradient of the loss with respect to each sample, $u_i$, without interference from other samples $u_{j \neq i}$**. In other words, **we can calculate the loss independently for each synthetic sample and update it using the derived gradient**. This explains why we do not need to factor in the batch size $B$ when considering our memory complexity. It also elucidates why our method offers a flexible trade-off between memory and runtime.
>
> In contrast, TESLA and all existing variations of MTT are unable to avoid the batch size in their memory complexity. This is because the **training trajectories are determined by batches of synthetic samples**, leading to the loss and updating gradient being inseparable for a single synthetic sample.

---

> ### Author Response · Authors · 2023-11-16
> **Response to Reviewer YDsU (2/3)**
>
> **Weakness 4: The authors should have more experiments on NAS. Because NAS is a practical application of dataset distillation, The experiments on Table 3 are not sufficient.**
>
> **Answer:**
>
> Thank you for your valuable suggestion. **In response, we conducted additional experiments for NAS (Neural Architecture Search) on CIFAR-100 and Tiny-ImageNet**. We randomly selected three groups of 100 networks from NAS-Bench-201 for these experiments. Specifically, we compared our methods with other baselines by training the 100 different architectural models on synthetic datasets with an IPC of 10 for 200 epochs. The table below presents the averaged Spearman's correlations between the rankings obtained from the synthetic datasets and the ground-truth rankings. The results clearly demonstrate that **our method continues to outperform other baselines on both evaluated NAS tasks**. We will include these results in our final print.
>
> |        | Random    | DM        | FrePo     | MTT       | **Ours**      |
> |--------|-----------|-----------|-----------|-----------|-----------|
> | CIFAR-100 | 0.15±0.03 | 0.11±0.05 | -0.11±0.08 | 0.39±0.03 | **0.61±0.06** |
> | Tiny   | 0.19±0.05 | 0.18±0.06 | -0.12±0.11 | 0.24±0.04 | **0.42±0.06** |
>
>
> **Question 1: The study of memory cost presented in Figure 3 could be more detailed. For instance, a comparison of complexity versus batch size or number of steps relative to the baseline TESLA would be informative.**
>
> **Answer:**
>
> We appreciate your valuable suggestion. **In fact, we offer a detailed comparison of our memory complexity concerning batch size and the number of steps relative to the baseline TESLA in the initial paragraph (Low Memory Complexity) of Section 3.3**. Due to space constraints, we chose not to redundantly include this information in the caption of Figure 3. Following your recommendation, we will consider adding a brief reference in the caption to assist readers in quickly locating this section.
>
>
> **Question 3: There seems to be a performance gap between the CNNs and the ViTs. Could authors have the cross-archi experiments on the distilled images trained with ViT. (Train on ViT and test on CNNs).**
>
> **Answer:**
>
> We appreciate your attention to the detail. This performance gap are generally exist in all the evaluated baseline method according to our reported results in Table 2. This is because all of the compared method (including ours) employ the ConvNet model as the proxy for generating synthetic samples. Due to the dissimilarity of the structures between ViT and ConvNet, all the evaluated results demonstrate a better generalization ability among models which also involving convolutional layers (e.g., ResNet18 and VGG) but exhibit a relatively poor performance on ViT.
>
> For your reference, **we conduct additional experiments of using ViT as the proxy model for DM, MTT and our method for distilling CIFAR-10 with an IPC of 50 and implement the cross-architecture experiments on randomly initialized ViT, ConvNet and ResNet18**. The results are presented in the following table. According to the results, although the results, obtained on ConvNet and ResNet 18, are remarkably lower than those obtained by using ConvNet as the proxy, **our method still exhibit the outstanding generalization performance among other baselines**. this observation underscores **the better ability of our method to capture essential cross-architecture features for persisting training effects**.
>
> |          |   DM    |   MTT   |   **Ours**   |
> |----------|---------|---------|----------|
> |   ViT    | 32.5±0.3| 36.8±0.6| **44.8±0.4** |
> | ConvNet  | 44.6±0.6| 51.2±0.4| **54.4±0.1** |
> | ResNet18 | 24.6±3.1| 51.8±0.8| **56.7±0.5** |

---

> ### Author Response · Authors · 2023-11-16
> **Response to Reviewer YDsU (3/3)**
>
> **Question 4: The SOTA of imageNet-1k should be SRe^2L [A] instead of TESLA. the claim of SOTA in imagenet-1k is not objective.**
>
> **Answer:**
>
> Thank you for your attention to detail. To the best of our knowledge, SRe^2L is developed for distilling data exclusively for ResNet models on the Tiny ImageNet and ImageNet-1k datasets. This specific application is somewhat because SRe^2L is a direct adaptation of model inversion and batch normalization statistics alignment techniques initially proposed for data-free knowledge distillation tasks [B].
>
> Notably, according to the experimental results reported by SRe^2L and its released code, we identify 3 reasons to not include SRe^2L in our formal evaluations: 1. the crafted data is shown to be effective primarily for training ResNet models with batch normalization layers, 2. SRe^2L is not effective for distilling other baseline datasets like CIFAR-10/100, and 3. **Most importantly**, when using synthetic data generated by SRe^2L for ImageNet-1k for model training, **it is necessary to involve CutMix data augmentation [C] to avoid a significant performance drop**.
> **This means that the amount of data actually used for model training is far greater than what is reported (e.g., IPC=50), and it also requires more epochs as well as training time for models to converge**.
>
> Given these limitations and the differing application scenarios, we believe it would not be fair to include SRe^2L in comparisons with other evaluated baselines that follow the same distillation protocols.
>
> [A] Zeyuan Yin, Eric Xing, and Zhiqiang Shen. Squeeze, recover and relabel: Dataset condensation at imagenet scale from a new perspective. In NeurIPS, 2023. 1, 2, 3, 4, 5,
>
> [B] Hongxu Yin, Pavlo Molchanov, José M. Álvarez, Zhizhong Li, Arun Mallya, Derek Hoiem, Niraj K. Jha, Jan Kautz: Dreaming to Distill: Data-Free Knowledge Transfer via DeepInversion. CVPR 2020: 8712-8721
>
> [C] Sangdoo Yun, Dongyoon Han, Sanghyuk Chun, Seong Joon Oh, Youngjoon Yoo, Junsuk Choe: CutMix: Regularization Strategy to Train Strong Classifiers With Localizable Features. ICCV 2019: 6022-6031
>
>
> **Weakness 3: The authors append some visualizations of the distilled images in the appendix. & Question 2: Visualization of the distilled ImageNet dataset.**
>
> **Answer:**
>
> Due to the limited elaboration, we found that we did not fully grasp the specific weaknesses and questions raised by the reviewer. If you consider these points to be significant weaknesses or questions, could you please provide additional details to assist us in providing a more thorough response?

---

### Official Review · Reviewer_oYNQ · 2023-10-31

**Soundness:** 2 fair
**Presentation:** 3 good
**Contribution:** 2 fair
**Rating:** 5
**Confidence:** 4

**Summary:**

This paper introduces a novel approach to dataset distillation, focusing on synthesizing smaller, condensed datasets while maintaining essential information.

The authors propose a method that minimizes the prediction discrepancy between models trained on a large original dataset and models trained on a small distilled dataset.

This is achieved through an adversarial framework, which distinguishes it from existing distillation methods that rely on nested optimization or long-range gradient unrolling.

**Strengths:**

Clarity and Readability: The paper is well-written and easy to understand, making it accessible to a wide audience.ch

Novelty: The use of an adversarial framework for dataset distillation is a somewhat new and innovative approach to the problem.

**Weaknesses:**

1. Efficiency Comparison: The paper highlights a significantly reduced storage requirement for the proposed method compared to MTT. It would be beneficial to include a comparative analysis of this result with other existing dataset distillation (DD) frameworks to provide a broader context and assess the method's efficiency against a wider range of approaches;


2. Scalability of the Proposed Method: While the proposed method offers a more efficient dataset distillation (DD) framework, it raises questions about its scalability. It would be valuable to see DD results using the original resolution of ImageNet, as the resolution setting is not explicitly mentioned in the paper (and it is assumed that the authors present results with reduced resolution). Additionally, it's advisable to include results on training models directly on ResNet rather than transferring to it, as there are existing DD methods that operate in this specific setting. Comparing against these methods would provide a more comprehensive assessment of the proposed method's scalability.

**Questions:**

1. Efficiency Comparison:

a. Can you provide a detailed efficiency comparison of your proposed method with other existing dataset distillation (DD) frameworks, particularly in terms of storage requirements?

b. How does your method compare in terms of efficiency when using original ImageNet image resolutions? Is the reduced resolution setting explicitly mentioned in the paper?

c. Could you consider providing results on training models directly on ResNet (without transfer learning), as some DD methods operate in this specific setting? What insights can you offer on the efficiency and performance of your method in this scenario?

2. Scalability of the Proposed Method:

a. Given that your method aims to offer a more efficient DD framework, what scalability challenges or considerations have you encountered when dealing with the original resolution of ImageNet images?

b. Can you elaborate on the choice of not providing results on models trained directly on ResNet, especially when other DD methods operate in this mode? What insights can you provide on the scalability and effectiveness of your approach under this setting?

---

> ### Author Response · Authors · 2023-11-16
> **Response to Reviewer oYNQ (1/3)**
>
> Thank you for recognizing the strengths of our paper. We are pleased that you found our paper well-written and clear, ensuring its accessibility to a broad audience. Your acknowledgement of the novelty of our approach, particularly the use of an adversarial framework for dataset distillation, is greatly appreciated. In response to the weaknesses and questions you've pointed out, we offer the following clarifications and arguments.
>
> **Question 1.a: Can you provide a detailed efficiency comparison of your proposed method with other existing dataset distillation (DD) frameworks, particularly in terms of storage requirements?**
>
> **Answer:**
>
> Thank you for your suggestion. To clarify, we would like to first emphasize that in the caption of Table 1, we explicitly noted that, apart from our method and TESLA, all other evaluated baseline methods have missing entries in Table 1's results. This omission is due to the fact that **these methods, when faced with a given computational resource (a server equipped with a TESLA A100), were unable to perform data distillation on the evaluated datasets under these specific scenarios**. We have elaborated extensively on the reasons behind the infeasibility of these methods for data distillation in these scenarios in Appendix A.2. It is important to note that due to the space limit of the paper, we specifically chose to highlight our method's efficiency improvements by comparing it to the only effective competitor, TESLA, using the challenging ImageNet-1k dataset, with a focus on memory usage and runtime.
>
> **Regarding storage requirements**, it is worth mentioning that **only our method and training trajectory matching-based methods (including MTT, FTD, and TESLA) have such demands**. This is because both our method and these MTT-based methods are grounded in the teacher-student paradigm, where imitation-based objectives are defined to transfer essential information from the original datasets to their distilled counterparts, preserving the training effect. However, **the effective utilization of information from the teacher model empowers our two frameworks to consistently outperform other methods in the majority of evaluated scenarios**. Furthermore, since we do not need to store snapshots of teacher models trained on the original data, **our method fundamentally reduces storage requirements to just $\frac{1}{epoch}$ compared to MTT-based methods**.

---

> ### Author Response · Authors · 2023-11-16
> **Response to Reviewer oYNQ (2/3)**
>
> **Question 1.c & 2.b: Could you consider providing results on training models directly on ResNet (without transfer learning), as some DD methods operate in this specific setting? What insights can you offer on the efficiency and performance of your method in this scenario? Can you elaborate on the choice of not providing results on models trained directly on ResNet, especially when other DD methods operate in this mode? What insights can you provide on the scalability and effectiveness of your approach under this setting?**
>
> **Answer:**
>
> Thank you for your attention to detail. To the best of our knowledge, we have **only identified one related work SRe^2L** [A], which was presented at NeurIPS 2023, **focusing on distilling data exclusively for ResNet models** on the Tiny ImageNet and ImageNet-1k datasets. This specific application is somewhat because SRe^2L is a direct adaptation of model inversion and batch normalization statistics alignment techniques initially proposed for data-free knowledge distillation tasks [B].
>
> Notably, according to the experimental results reported by SRe^2L and its released code, **we identify 3 reasons to not include SRe^2L in our formal evaluations**: 1. the crafted data is shown to be effective primarily for training ResNet models with batch normalization layers, 2. SRe^2L is not effective for distilling other baseline datasets like CIFAR-10/100, and 3. **Most importantly**, when using synthetic data generated by SRe^2L for ImageNet-1k for model training, **it is necessary to involve CutMix data augmentation [B] to avoid a significant performance drop**.
> **This means that the amount of data actually used for model training is far greater than what is reported (e.g., IPC=50), and it also requires more epochs as well as training time for models to converge**.
>
>
> Given these limitations and the differing application scenarios, we believe it would not be fair to include SRe^2L in comparisons with other evaluated baselines that follow similar distillation protocols.
>
> Additionally, we want to emphasize that **in Appendix A.4, we conducted a study using ResNet18 as a proxy model to generate distilled data for CIFAR-10 and Tiny ImageNet and evaluate their generalization ability on various architectural models including ResNet18**. The experimental results indicate that **our method outperforms the results reported by SRe^2L when training ResNet18**. For your reference, we **conducted additional experiments using ResNet18 as the proxy to generate distilled data for CIFAR-100 and ImageNet-1k (224x224)**, with a distillation budget set to IPC=50 for all four benchmark datasets. The results are presented in the table below. We can see from the results that our method performs slightly worse than SRe^2L on ImageNet-1k but remarkably outperforms SRe^2L on CIFAR-10/100 and Tiny ImageNet.
>
> |         | CIFAR-10 | CIFAR-100 | Tiny ImageNet | ImageNet-1k |
> |---------|----------|-----------|---------------|-------------|
> | SRe^2L  | 52.8     | 35.8      | 41.1          | 46.8        |
> | Ours    | 63.2     | 59.4      | 54.8          | 43.5        |
>
> [A] Zeyuan Yin, Eric Xing, Zhiqiang Shen: Squeeze, Recover and Relabel: Dataset Condensation at ImageNet Scale From A New Perspective. NeurIPS 2023
>
> [B] Hongxu Yin, Pavlo Molchanov, José M. Álvarez, Zhizhong Li, Arun Mallya, Derek Hoiem, Niraj K. Jha, Jan Kautz: Dreaming to Distill: Data-Free Knowledge Transfer via DeepInversion. CVPR 2020: 8712-8721
>
> [C] Sangdoo Yun, Dongyoon Han, Sanghyuk Chun, Seong Joon Oh, Youngjoon Yoo, Junsuk Choe: CutMix: Regularization Strategy to Train Strong Classifiers With Localizable Features. ICCV 2019: 6022-6031

---

> ### Author Response · Authors · 2023-11-16
> **Response to Reviewer oYNQ (3/3)**
>
> **Question 1.b & 2.a: How does your method compare in terms of efficiency when using original ImageNet image resolutions? Is the reduced resolution setting explicitly mentioned in the paper? Given that your method aims to offer a more efficient DD framework, what scalability challenges or considerations have you encountered when dealing with the original resolution of ImageNet images?**
>
> **Answer:**
>
> Thank you for your attention to detail. In fact, **at the beginning of Section 4.1**, titled "Experimental Setups," **we explicitly state that we adhere to the protocol of previous arts (FRePo and TESLA) by resizing ImageNet-1K images to 64 × 64 resolution in our evaluations to ensure fair comparisons**.
>
> Based on the analysis and experimental results we provided in response to the previous question, our method demonstrates **the capability to use the ResNet18 model as the proxy for distilling ImageNet-1k in its original resolution**. Additionally, we want to reiterate that, as per our memory complexity analysis in Section 3.3, our method, along with SRe^2L, can independently calculate and update the loss for each synthetic data point. Our memory complexity is primarily dominated by the partial derivative of logits difference between a teacher and a student for a single synthetic sample, which involves only a trivial backpropagation through the two models. Mathematically, this leads to our method having a memory complexity that is only twice that of SRe^2L. However, while TESLA is the only generally defined dataset distillation method that can be applied to distil 64x64-sized ImageNet-1k (when IPC is larger than 10), attempting to use it for the original resolution ImageNet-1k leads to out-of-memory (OOM) errors on our TESLA A100 server

---

### Official Review · Reviewer_Q2TW · 2023-11-10

**Soundness:** 2 fair
**Presentation:** 3 good
**Contribution:** 2 fair
**Rating:** 5
**Confidence:** 4

**Summary:**

This paper proposes a new dataset distillation technique that can (1) save the computational cost compared to several baselines and (2) improve the test accuracy when training new networks. The framework utilizes the idea of adversarial training that tries to maximize the disagreement of student and teacher networks on the synthetic samples. Several approximation techniques are introduced to make the optimization practical and efficient. Experiments show that the proposed method can outperform baseline methods.

**Strengths:**

- The topic is timely as the dataset sizes are getting larger.
- The authors have conducted a lot of experiments on various tasks and datasets.
- The writing quality is high and the presentation is clear.

**Weaknesses:**

- Why having a form of Equation (3) to train the synthetic samples?  I understand it might become easier to split the loss into two components if using Equation (3), and also it might have a similar form with Equation (1), but taking the logarithm over the loss values still look uncommon to me.
- How to verify that the synthetic samples are really approaching “hard samples”? This is one important assumption but I cannot find a verification for this. Also, it seems that $x_e$ depends on $\theta_e^S$, which means that they could be different samples over time. Therefore, why would a fix single set of $u$ can approach this dynamic set of "hard samples"? This is the main difficulty I have when I try to understand why the proposed method would work.
- I noticed that the authors use soft label instead of one-hot label. However, some baseline methods in Table 1 and Table 2 do not apply this. I think it would be beneficial to indicate this point when comparing with other baselines.
- Figure 5(b): it seems that the performance will drop when using more checkpoints, which looks counterintuitive to me: the approximation outperforms the original loss objective in terms of the final performance. It would be beneficial to provide more analysis on this point.

**Questions:**

See the above section. I am open to change my score based on the authors' responses.

---

> ### Author Response · Authors · 2023-11-16
> **Response to Reviewer Q2TW (1/3)**
>
> Thank you for acknowledging our paper's strengths, including the relevance of our topic in the era of large datasets, the breadth of our experiments, and the clarity of our presentation. We really appreciate your conscientious and responsible review, detailed comments and insightful questions. Our responses are as follows.
>
> **Weakness 2: How to verify that the synthetic samples can approach “hard samples”? why would a fixed single set of can approach dynamic sets of "hard samples"? why does the proposed method work?**
>
> **Answer:**
>
> Thank you for your insightful questions and the attention to detail! We appreciate the opportunity to clarify the rationale behind our methodology. In the following context, we first argue **the effectiveness of using "hard samples" as distilled data** and discuss **why we use fixed distilled data to target common "hard samples" for all snapshots simultaneously**. Then, we **empirically validate that our crafted data qualifies the defined "hard samples"**.
>
> Hard samples, as defined in Section 3.2, meet two criteria for a snapshot $\theta_e^\mathcal{S}$ of a proxy student model: 1. High class probability of $p(x|y)$ (substitutable by $p(y|x,\theta^{\mathcal{T}})$), and 2. A significant logits difference between $f \theta^\mathcal{T}$ and $f_{\theta_e}^\mathcal{S}$. We consider an intuitive scenario where the distilled dataset $\mathcal{S}$ is a smaller-scale sample from the original $\mathcal{T}$. Empirically, each $f_{\theta_e^\mathcal{S}}$ only has similar predictions to $f_{\theta^\mathcal{T}}$ for samples contained in the current $\mathcal{S}$ or those with similar features to the current distilled data. This implies the existence of "hard" samples in the original data distribution, **identifiable by the teacher but not the student due to a significant gap of contained features between $\mathcal{S}$ and $\mathcal{T}$**. Our adversarial scheme reduces this gap, **encouraging the exploration of hard samples and gathering informative features acknowledged by the teacher but absent in $\mathcal{S}$**.
>
> Additionally, **to avoid inductive bias towards any specific training stage (we already emphasized its harm in Figure 2)**, we construct our objective in Equation (3) as **maximizing the summation of logits difference over all snapshots** instead of **maximizing the logits difference for each snapshot independently**. This also explains why we use fixed synthetic data to target common "hard samples" for all snapshots simultaneously.
>
> Next, we experiment on CIFAR-10 to verify that crafted distilled data can meet our definition of hard samples. We first create an initial distilled dataset $\mathcal{S}^0=\\{(u^0, f_{\theta^\mathcal{T}}(u^0))\\}$ with an IPC of 50 and gather a set of $\\{f_{\theta_e^{\mathcal{S}^0}}\\}$ with $10$ select snapshots, collected at regular intervals during training. Post application of our method, we obtained a crafted dataset $\mathcal{S}^*$. In comparing the logits of $f_{\theta^\mathcal{T}}$ and $\\{f_{\theta_e^{\mathcal{S}^0}}\\}$ over $u^0$ and $u^*$, we observe that compared to the logits predicted over $u^0$, 1.  the logits for the corresponding class (the original ground truth class) of $f_{\theta^\mathcal{T}}$ over  $u^*$ significantly increase (averaging from 8.5 to 16.2) and decrease for non-corresponding classes (averaging from -1.0 to -2.2); 2. conversely, $\\{f_{\theta_e^{\mathcal{S}^0}}\\}$'s averaged logits over $u_*$ decrease on the corresponding class (averaging from 7.8 to 3.4), and increase for non-corresponding classes (averaging from -1.0 to -0.5). These empirical observations demonstrate that the crafted distilled data **increase the probability of $p(y_u|u,\theta^\mathcal{T})$ and enlarge the logits difference**. Namely, our adversarial scheme effectively **transforms the randomly sampled $u_0$ into the defined hard samples for $\\{f_{\theta_e^{\mathcal{S}^0}}\\}$ by injecting informative features that are identifiable with teachers but absent in $\mathcal{S}^0$**. The performance improvement of $\mathcal{S}^*$ (i.e., 75.0) over $\mathcal{S}^0$ (i.e., 60.5) also confirms the benefit of the features introduced by our adversarial scheme for model training.

---

> ### Author Response · Authors · 2023-11-16
> **Response to Reviewer Q2TW (2/3)**
>
> **Weakness 1: Why have a form of Equation (3) to train the synthetic samples? I understand it might become easier to split the loss into two components if using Equation (3), and it might have a similar form with Equation (1), but taking the logarithm over the loss values still looks uncommon to me.**
>
> **Answer:**
>
> We greatly appreciate the thorough review by the reviewer and the opportunity to clarify the benefits of taking the logarithm of the probability-weighted logits difference in Equation (3). As mentioned in our response to Weakness 2 above, we formulate the objective for crafting our synthetic data as Equation (3) to simultaneously create "hard samples" that satisfy our defined conditions for all involved training snapshots (checkpoints). This formulation helps prevent inductive bias towards any specific training stage.
>
> Taking the logarithm of the term $p(u|y_{u}) d\left( f_{\theta^{\mathcal{T}}}\left(u\right) , f_{\theta_e^{\mathcal{S}}}\left(u\right) \right)$ in this equation offers **two main advantages**. First, as you've noted, it **decouples the maximization of logits difference in Equation (4) from the maximization of $p(u|y_{u})$ (substitutable by $p(y_u|u,\theta^{\mathcal{T}})$)**, allowing us to flexibly tradeoff optimization between the two by introducing an additional hyperparameter $\alpha$. We also discuss its effectiveness in Appendix A.5.
>
> Second, leveraging the properties of the logarithmic function and the chain rule, we can express the derivative of $\ln(A(u))$ w.r.t u as $\frac{\partial}{\partial u}[\ln(A(u))] = \frac{1}{A(u)} \cdot \frac{\partial A(u)}{\partial u}$. Compared to directly taking the derivative of $A(u)$ w.r.t u, taking the logarithm of $A(u)$ and then obtaining the derivative naturally weights it by the value of $A(u)$. **This trick is widely applied in scenarios where gradients need to be weighted by the value of the function itself**. For instance, in the design of loss functions, applying a logarithmic transformation to the output (e.g., taking the log-likelihood loss) can alter the gradient scale, contributing to the stability of the training process. In the loss function defined in Equation (4), **taking the logarithm of logits difference can also automatically enlarge the gradient for samples with smaller differences**.
>
>
> **Weakness 3: I noticed that the authors use soft label instead of one-hot label. However, some baseline methods in Table 1 and Table 2 do not apply this. I think it would be beneficial to indicate this point when comparing with other baselines.**
>
> **Answer:**
>
> Thank you for your valuable suggestion. Regarding Tables 1 and 2, the results for all baseline methods are accurately derived by adhering to their specific default protocols. Among them, FRePo and RCIG utilize optimal soft labels, which they have learned, to create their distilled dataset. In contrast, other methods use standard one-hot labels. We will make this distinction clear in our final manuscript.
>
> Moreover, to convincingly demonstrate that our synthetic images encapsulate the most informative features essential for achieving our prediction-matching goal and maintaining the training effect, we **reconstruct distilled datasets for DM, MTT, and FTD by substituting one-hot hard labels with soft labels (logits) output by our pretrained teacher models**. We then train models using our adopted $L1$ loss. The following table presents the test accuracies (%) of ConvNet models trained on these logits-labeled distilled datasets, along with a comparison of the variations relative to the results in Table 1. The results demonstrate that **our methods still outperform other baselines in all the evaluated scenarios**. This observation emphasizes that **our synthetic images are effective for incorporating the essential features that are critical for meeting our prediction-matching objective** and **can better utilize teachers' logits for preserving the training effect**.
>
> | Dataset         | DM               | MTT              | FTD              | **Ours**             |
> |-----------------|------------------|------------------|------------------|------------------|
> | CIFAR10-ipc 50  | 60.7±0.2 ↓2.3    | 71.1±0.4 ↓0.5    | 73.4±0.3 ↓0.4    | **75.0±0.2**         |
> | CIFAR10-ipc 500 | 73.8±0.2 ↓0.5    | 79.3±0.2 ↑0.7    | 79.8±0.4 ↑1.1    | **83.4±0.3**         |
> | CIFAR100-ipc 10 | 23.2±0.5 ↓6.5    | 39.2±0.5 ↓0.9    | 43.2±0.2 ↓0.2    | **44.6±0.3**         |
> | CIFAR100-ipc 50 | 44.5±0.1 ↑0.9    | 49.1±0.3 ↑1.4    | 51.1±0.3 ↑0.4    | **53.3±0.2**         |
> | Tiny-ipc 10     | 15.8±0.3 ↑2.9    | 24.2±0.3 ↑1.0    | 25.4±0.3 ↑0.9    | **30.0±0.3**         |
> | Tiny-ipc 50     | 23.5±0.3 ↑2.3    | 31.5±0.2 ↑3.5    | 35.5±0.3 ↑4.0    | **38.2±0.4**         |

---

> ### Author Response · Authors · 2023-11-16
> **Response to Reviewer Q2TW (3/3)**
>
> **Weakness 4: Figure 5(b): it seems that the performance will drop when using more checkpoints, which looks counterintuitive to me: the approximation outperforms the original loss objective in terms of the final performance.**
>
> **Answer:**
>
> We really appreciate your attention to detail and pointing out this counterintuition! We carefully studied this phenomenon and found out that **this slight performance drop is attributed to that we did not correspondingly tune the learning rate used for updating the synthetic when deriving the results of involving more than five checkpoints for calculating the loss**. That is, we maintained to use the learning rate initially set for involving five checkpoints. However, our defined loss function calculates the summation of the logits differences caused by synthetic samples between the teacher and multiple checkpoints, rather than the average. Therefore, introducing additional checkpoints alters the final loss, necessitating an adjustment in the learning rate for updating the synthetic data. Following this analysis, we carefully **recalibrated the learning rate across various scenarios involving different numbers of checkpoints, retested the performances, and reported the results in the table below**. The experimental outcomes remain consistent with our original paper's assertion: setting the number of involved checkpoints higher than five does not lead to significant performance changes. We greatly appreciate your attention to the detail of our work and will correct the Figure 5 (b) in our final print.
>
> | checkpoint    | 1           | 3           | 7           | 10          | 12          | 15          | 20          |
> |---------------|-------------|-------------|-------------|-------------|-------------|-------------|-------------|
> | Acc (%)       | 67.8±0.5    | 74.2±0.3    | 74.9±0.4    | 75.1±0.2    | 75.0±0.1    | 75.3±0.2    | 75.1±0.3    |

---

### Author Response · Authors · 2023-11-21

We appreciate all the reviewers for your valuable feedback! In response. we have thoughtfully considered and addressed each of your points, incorporating additional experiments where necessary. We believe these responses further deepened the understanding and highlighted the contributions of our work. Should there be any more concerns or questions affecting your review, we are ready to further discuss or provide additional clarifications. We eagerly await your feedback.

---

### Author Response · Authors · 2023-11-23
**Distinctiveness and Contributions of Our Work for the Dataset Distillation Community**

Here, we would like to highlight the **distinctiveness** and **contributions** of our work for the dataset distillation community:

1. Our adversarial prediction-matching framework **pioneers a novel perspective for the dataset distillation techniques**. Existing methods, relying on surrogate objectives such as gradient-matching or training trajectory matching, encounter inefficiency due to the absence of direct feedback on the effectiveness of the current distilled data. This results in updates to distilled data that cannot be directly aligned with improving the training effect. In contrast, our adversarial prediction-matching scheme **effectively narrows the feature gap between the distilled and original datasets by utilizing the feedback of prediction disagreement** between proxy teacher models (trained on the original dataset) and proxy student models (trained on the distilled dataset) as a conduit.

2. We demonstrated that our defined loss function used to update synthetic data hinges on **a trivial single-level optimization**. Importantly, the memory complexity required to compute the gradient updates for synthetic data is proven to be significantly lower than that of the most memory-efficient MTT-based method, TESLA. These characteristics enable our method to achieve superior outcomes with only 2.5× less memory and 5× reduced runtime compared to TESLA when distilling the ImageNet-1K dataset.

3. Following the standard evaluation protocol, our method demonstrated **remarkably superior performance across multiple benchmark datasets compared to baseline methods in all evaluated scenarios**. Additionally, our approach showed enhanced cross-architecture generalization performance. The practicality of our method is further supported by our NAS experiments, where it significantly outperformed other baseline methods.

---

### Author Response · Authors · 2023-11-23
**Rebuttal Summary and Appeal for Responsible Assessment**

Because we did not receive any feedback from reviewers for the rebuttal during the discussion, we compiled a summary to facilitate the later discussions between the Area Chair and reviewers. This summary encompasses **the reviewers' acknowledgments, concerns, and our respective responses**. We would like to respectfully convey our appreciation for the equitable, thorough, and transparent scholarly atmosphere at ICLR, and earnestly hope that our submission will be accorded **a responsible and equitable assessment**.

We are grateful for all the reviewers' acknowledgment of **our adversarial prediction-matching framework as an innovative approach in dataset distillation**. Their commendation of the paper's **writing quality** and **readability** is also highly valued. We thank reviewers Q2TW and YDsU for acknowledging **the superiority of our experimental results over other evaluated baseline methods**. Reviewer YDsU's acknowledgment of our method's **effectiveness in addressing two challenges of dataset distillation including memory issues and training costs** is also appreciated.

Regarding concerns, we noted that **there was not a common weakness or limitation in our method identified by all reviewers**. Each reviewer seems to present distinct concerns and questions.

Reviewer **Q2TW** questioned **why we use a logarithmic function for loss values in Equation (3) and how to verify that the synthetic samples are indeed "hard samples" as our defined**. In response, we clarified that **logarithmic application has two main advantages**: decoupling the maximization of logits difference from probability and leveraging the logarithmic function's intrinsic property to automatically adjust gradient magnitude by the loss values. We also **argued the effectiveness of using "hard samples" associated with an intuitive experiment as evidence**. Additionally, we acknowledged the suggestion to specify whether the evaluated baseline methods use one-hot or soft labels in the paper. We also have further investigated and explained the counterintuitive performance drop seen in Figure 5(b) when using more checkpoints and provided revised results.

Reviewer **oYNQ**'s comment focused on the need for additional empirical comparisons of 1. **storage requirements** between our method and other baselines and, 2. **the efficacy of using ResNet as a proxy model and on ImageNet-1K at its original resolution**. In response to the first concern, we clarify that **only our method and training trajectory matching (MTT) based approaches require storage** for teacher snapshots. However, our **effective utilization of information from the teacher model** enables our method to consistently outperform other methods in all evaluated scenarios with **only $\frac{1}{epoch}$ storge than MTT-based methods**. Regarding the second concern, we first highlighted that we have studied the performance of using ResNet18 as a proxy model for distilling CIFAR-10 and Tiny ImageNet in Appendix A.4. For the reviewer's reference, we also provide additional experiments with ResNet18 on CIFAR-100 and ImageNet-1K (224*224), demonstrating our method's effectiveness.

For reviewer **YDsU**'s questions, we first **delineated our method's distinction from the earliest dataset distillation method DD** [A] from three aspects. We then clarified **why our method’s memory complexity does not rely on batch size counting like MTT-based approaches**. Further, we conducted additional experiments, as suggested, including **NAS tests** on CIFAR-100 and Tiny ImageNet and **using ViT as a proxy for distilling datasets** and evaluated them over CNN models. Finally, we presented **three justifications for excluding SRe^2L** [B] (a recent dataset distillation work tailored for particular models with batch normalization, accepted by NeurIPS 2023) from our baseline methods under evaluation.

[A] Tongzhou Wang, Jun-Yan Zhu, Antonio Torralba, and Alexei A. Efros. Dataset distillation. CoRR, abs/1811.10959, 2018.

[B] Zeyuan Yin, Eric Xing, Zhiqiang Shen: Squeeze, Recover and Relabel: Dataset Condensation at ImageNet Scale From A New Perspective. NeurIPS 2023

---

### Meta-Review · Area_Chair_fXDq · 2023-12-18

**Metareview:**

This paper proposes a new method for dataset distillation -- the problem of condensing a dataset to a smaller size while minimizing the loss of predictive performance of models trained on it. The key idea behind this method is to directly minimize the discrepancy between predictions on the real data distribution of two models, one trained on the original dataset and one trained on the distilled one. This discrepancy is operationalized through an adversarial loss.

The reviewers unanimously praised the paper's clarity/readability and simplicity, but expressed reservations about the fairness of experimental comparison, the novel contribution with respect to other similar methods, and the scalability of the proposed approach. Some of these concerns were addressed in the authors' response. Unfortunately, the reviewers did not acknowledge or further engage with the authors' response. Based on this, I decided to down-weight the numerical scores they provided, and read and review the paper myself, taking into account all reviews and responses.

After completing this process, I have come to the conclusion that although this paper provides some interesting results and insights, it is below the bar of what would be expected of an ICLR paper in terms of methodological and/or experimental contributions, for the following reasons:
- **Limited methodological contribution**. As pointed out by many reviewers, there is very little in the way of methods/algorithmic contributions in this paper. The approach boils down to a prediction matching approach, that has been widely studied in the *model* distillation literature. Surprisingly, some of the insights that emerged from that line of work (e.g., the importance of soft labels) are not fully internalized in this work, leading to some issues with evaluation (see point below).
- **Exaggerated claims of novelty / missing related work**. The paper claims, at various points, to offer a "new perspective" on dataset distillation based on minimizing prediction discrepancy based on imitation/prediction-matching/adversarial training. This, however, ignores a long line of DD work that uses prediction matching (Bohdal et al. 2020), feature matching (Wang et al. 2022) and/or adversarial objectives (Zhao & Blien, 2022a, b). While non of these is identical, they all share the premise of directly matching the posteriors P(Y|X) for different models trained on the full and distilled datasets, which makes the claims of a "new perspective" of this paper seem overblown.
- **Missing baselines**. The authors justify not including Dataset Distillation [1] as a baseline due to computational limitations, but given the proximity of this method to DD, it seems almost necessary to do so, even if in smaller/lower-resolution settings, to truly understand the advantage of the proposed objective over the DD variant.
- **Lack of ablation with respect to hard/soft labels**. As pointed out by reviewer Q2TW, the original experimental framework does not control for whether the samples are distilled with hard or soft labels, using different regimes for each method/baseline. This is problematic because there is mounting evidence in recent work that soft labels might be responsible for much of the improvement seen in DD methods that use it. Although the authors provided additional results where they claim to re-run baselines with soft labels, I am skeptical of those results since they don't seem to significantly improve (and in many cases, degrade) the performance of the baselines. I have seen other results for comparing of soft vs hard labels in DD, and in all cases the differences are much higher. To be clear, I don't mean to imply wrong doing by the authors, instead, I suspect there might be a bug or hyperparameter issue that is limiting the efficacy of soft labels for the baselines.

Overall, I think this is a borderline paper, that is missing important components that would justify recommending acceptance.

* Bohdal, Yang, and Hospedales. "Flexible Dataset DIstillation: Learn Labels Instead of Images", 2020.
* Wang, Zhao, Peng et al, "Cafe: Learning to condense dataset by aligning features.", CVPR 2022.
* Zhao & Bilen, "Dataset Condensation with Distribution Matching", 2022a.
* Zhao & Bilen, "Synthesizing Informative Training Samples with GAN", 2022b.
* Zhao, Li, Qin, Yu, "Improved distribution matching for dataset condensation", CVPR 2023.

**Justification For Why Not Higher Score:**

Limited novel contribution, experimental evaluation is not rigorous enough.

**Justification For Why Not Lower Score:**

N/A

---

### Decision · Program_Chairs · 2024-01-16

Reject